# Monoamine Neurotransmitters Control Basic Emotions and Affect Major Depressive Disorders

**DOI:** 10.3390/ph15101203

**Published:** 2022-09-28

**Authors:** Yao Jiang, Di Zou, Yumeng Li, Simeng Gu, Jie Dong, Xianjun Ma, Shijun Xu, Fushun Wang, Jason H. Huang

**Affiliations:** 1Institute of Brain and Psychological Sciences, Sichuan Normal University, Chengdu 610066, China; 2Department of Psychology, Medical School, Jiangsu University, Zhenjiang 210023, China; 3Department of Neurology, Affiliated Lianyungang Hospital of Chinese Medicine, Nanjing University of Chinese Medicine, Nanjing 222000, China; 4School of Pharmacy, Chengdu University of Traditional Chinese Medicine, Chengdu 616000, China; 5Department of Neurosurgery, Baylor Scott & White Health, Temple, TX 79409, USA; 6Department of Surgery, College of Medicine, Texas A&M University, Temple, TX 79409, USA

**Keywords:** astrocytes, emotional arousal, basic emotions, three primary color, monoamine, major depressive disorders

## Abstract

Major depressive disorder (MDD) is a common and complex mental disorder, that adversely impacts an individual’s quality of life, but its diagnosis and treatment are not accurately executed and a symptom-based approach is utilized in most cases, due to the lack of precise knowledge regarding the pathophysiology. So far, the first-line treatments are still based on monoamine neurotransmitters. Even though there is a lot of progress in this field, the mechanisms seem to get more and more confusing, and the treatment is also getting more and more controversial. In this study, we try to review the broad advances of monoamine neurotransmitters in the field of MDD, and update its effects in many advanced neuroscience studies. We still propose the monoamine hypothesis but paid special attention to their effects on the new pathways for MDD, such as inflammation, oxidative stress, neurotrophins, and neurogenesis, especially in the glial cells, which have recently been found to play an important role in many neurodegenerative disorders, including MDD. In addition, we will extend the monoamine hypothesis to basic emotions; as suggested in our previous reports, the three monoamine neurotransmitters play different roles in emotions: dopamine—joy, norepinephrine—fear (anger), serotonins—disgust (sadness). Above all, this paper tries to give a full picture of the relationship between the MDD and the monoamine neurotransmitters such as DA, NE, and 5-HT, as well as their contributions to the Three Primary Color Model of Basic Emotions (joy, fear, and disgust). This is done by explaining the contribution of the monoamine from many sides for MDD, such the digestive tract, astrocytes, microglial, and others, and very briefly addressing the potential of monoamine neurotransmitters as a therapeutic approach for MDD patients and also the reasons for its limited clinical efficacy, side effects, and delayed onset of action. We hope this review might offer new pharmacological management of MDD.

## 1. Introduction

Major depressive disorder (MDD) is considered a serious public health issue and is one of the leading causes of disability, disease burden and suicide deaths worldwide [1,2]. Even though there has been a lot of progress in this field, its mechanisms seem to get more and more confusing, and the treatment is getting more and more controversial [3]. Currently, its diagnosis and treatment are not being accurately executed, and a symptom-based approach is utilized in most cases due to a lack of precise knowledge regarding the pathophysiology. So far, the first-line treatment is still based on monoamine neurotransmitters or most antidepressant drugs are still targeting the monoamine system (including serotonin, noradrenaline, and dopamine), which has been showed to be the substrate for emotions ever since the 1950s–1960s in the last century [4,5,6,7].

Rather than solely focusing on symptomology, it is imperative to understand the underlying causal mechanisms. Recent research has shown that the processing of information and regulation of emotions are altered in MDD [3]. Indeed, MDD is a kind of affective disorder characterized by a persistent depressive mood and a lack of interest or pleasure [8]. However, even though MDD is a kind of emotional disorder, we know almost nothing about the emotional mechanisms or psychological mechanisms of MDD. Thus, the current etiological aspects of MDD fail to explain many fundamental questions about MDD; for example, we do not know why depression tends to occur in some vulnerable populations, such as adolescents, women, and the elderly.

Currently, many depressed patients become insensitive to medication and even further exhibit refractory depression [9,10]. Because there is no systematic theoretical guidance, the diagnosis and treatment often face a dilemma [11]; antidepressant treatment options are limited and, in some cases, ineffective [12]. In our previous studies, we have proposed a hypothesis of mood alterations in depression from the basic emotional aspects [13,14]. In this paper, we reviewed recent advances for monoamine neurotransmitters in MDD and further explored their roles in the emotional mechanisms of MDD, hoping to provide a new perspective for the neural mechanisms of depression.

## 2. Monoamine Neurotransmitters Mediate Three Core Effects

Emotion plays a very important role in our daily lives, affecting our social interactions, with more than 80–90% of human psychological problems being emotional problems. However, emotion is considered to be one of the earliest studied but least known subjects among all life sciences. Recently, many papers have linked emotions to monoamine neurotransmitters, such as the “new three-dimensional model”, which suggested that emotions are mediated by three monoamine neurotransmitters, including (norepinephrine, dopamine, and serotonin). In addition, these three monoamines make up the three dimensions for the emotional cubic [6]. However, even though tons of studies support the role of monoamines in emotions, their effects are quite mixed. For examples, most psychological stimulants or antidepressants are targeting all three monoamines and are used for almost all of the psychological disorders such as depression and anxiety. Here, we tried to differentiate the roles of monoamines into basic emotions, and originally proposed the ‘*Three Primary Color Model of Basic Emotions*’, which hypothesized that human beings have three core affects, including reward (joy), punishment (disgust), and stress (fear and anger) [13,15,16]. The monoamines work together to make different emotions, like the three primary colors (Figure 1). The dopamine (DA) system is involved in reward (joy), and the serotonin (5-HT) system is related to punishment (disgust or sadness), while noradrenaline (NE) is related to fear (anger) and the “fight or flight” behavior during stressful events. These three core affects might be called three prototypical emotions, which can be combined into a variety of compound emotions.

### 2.1. The Three Monoamines and Core Affects

The most important feature of depression is the absence of joyful emotions. Dopamine (DA) has become a synonym for joy and reward [17], thus increasing DA might be a good way for treating MDD [18,19], and, indeed, the US FDA (food and drug agency) recently approved ketamine as a drug capable of rapid antidepressant function in 2019 [20,21]. Ketamine itself is a drug and a stimulant, and one of its main functions is to increase release of dopamine. Even though many studies reported about the mechanisms of ketamine as antidepressants, ranging from effects on NMDA receptors to effects on GABA receptors [22,23], the major reason for ketamine to exert its antidepressant effect is related to DA release, possibly by activating astrocytic Ca^2+^ signaling and changing the bursting of neurons in the Habula nucleus [24]. Thus, raising brain dopamine is certainly a quick way to change the depressed mood [25].

Norepinephrine (NE) has been known as the neurotransmitter for emotional arousal [26], and its major function is inducing “fight or flight” behavior [27,28,29,30], and the “fear and anger” emotion [31]. Epinephrine and NE are secreted by the LC (Locus Coeruleus) in the brain, while they are released to the blood by adrenal glands. It was in 1923 that Cannon suggested that the major function of the NE is inducing stress in the body, leading to ‘fight or flight’ behavior. We proposed for the first time that anger and fear might be twin emotions as they share the same neurotransmitter NE/ adrenaline [32]. Indeed, some previous reports also suggested that NE and adrenaline mediated fear and anger; for example, Kabitzke et al tried to distinguish fear and anger based on these two neurotransmitters, and suggested that NE mediates fear and adrenaline mediates anger [33,34]. Here we suggested that fear and anger are two faces of the same coin, and mediated by the same neurotransmitters, together with the sympathetic nervous system [35].

Disgust is the least studied basic emotions; recently, a growing number of studies suggest that disgust plays a significant role in our lives and in many psychological disorders [35,36]. Many experiments have confirmed the reward function of DA and the mood arousal function of NE, but the relationship between serotonin (5-HT) and depression is much more important, for most current antidepressants are targeting 5-HT [37,38,39]. However, the role of 5-HT in depression is complicated by a diversity of 5-HT receptors, and the prevalence of its distribution in the organism. 5-HT has a wide range of receptors, including 14 different receptor subtypes, and are distributed throughout the body, especially in the gastrointestinal tract [40]. More than 90% of the 5-HT in the body are released in the gastrointestinal tract, and many 5-HT receptors are also distributed in the intestine, such as 5-HT1_A_ [41]. 5-HT in the gut produces a possibility that gut alterations may be important in the pathophysiology of MDD [42]. This may be the reason that many studies have recently found that the brain–gut axis alternation is an important cause of depression, and indeed the greatest side effect of increased 5-HT is an increase in gastrointestinal nausea and vomiting response [43].

In the central nervous system, 5-HT is mainly released from the Raphe nucleus, whose major function is sleep, so the main side effect is to increase the patient’s sleep, instead of making the patient excited [44]. Even though many papers proposed the antidepressant function of 5-HT, the real function of central 5-HT in treating MDD might be inducing sleep, sedation, and inhibition of the compulsory thought [45]. Thus, the function of 5-HT might be suppressing the body’s excitability, instead of being an antidepressant. Indeed, the increase of 5-HT induces a sedative response in the body, which is a behavioral inhibition process [46], similar to the prolonged helpless state during depression [47]. Indeed, recent emphasis has been focused mainly on the effects of monoamines on behavioral inhibition [48]. In addition, recent studies found that the majority of psychological problems stem from the patient’s disgust with the people around them [47]. 

### 2.2. The Significance of Basic Emotion Theory

The hypothesis of monoamine transmitters as the neural basis for emotion was first proposed in the 1950s [5], and it is still used as the basis for developing antidepressants. However, it is not clear how monoamine transmitters affect emotions in MDD patients. Basic Emotion Theory (BET) is the most prevalent theory about emotion, which suggests that human emotions are composed of a limited number of basic emotions. The basic emotions are evolved to accomplish fundamental life tasks, and each primary emotion has a unique neural structure and physiological basis. Although most studies agreed upon the idea that emotions may consist of some discrete emotions, there is no consensus on the number of basic emotions. For example, Ekman proposed that there are seven basic emotions, including fear, anger, joy, sadness, disgust, and contempt [49]. The seven basic emotions proposed by Ekman is like the 7-color disk, or like Newton’s disk (Figure 1A,B), but there are only three primary colors. Recently, we redefined the basic emotion theory and made an analogy between emotion and colors [15]. We suggested that human beings may have only three basic emotions, and we originally proposed the ‘Three Primary Color Model of Basic Emotions’. To differentiate them with the definition of basic emotions in previous studies [50,51,52], we called them core affects, including reward (joy), punishment (disgust), and stress (fear and anger) (Figure 1C) [13,15,16]. These three core affects can be combined into a variety of compound emotions.

The Basic Emotion Theory (BET) dates back to Darwin’s book The Expressions of the Emotion in Man and Animals in 1872, in which he proposed that even insects have basic emotions which are similar to those of humans [15,53]. However, even though the brain structures of insects and humans are very different, both insects and humans have three monoamine neurotransmitters: DA, NE, and 5-HT [15]. Thus, we suggested that these three monoamine neurotransmitters are the neural basis for three core emotions [16,31,54]. The three primary color model of basic emotions fully supports the theory of monoamine transmitters for MDD (Figure 1). Anyway, it is not clear what the role of these three core emotions and three monoamines is in the development of depression in addition to how depressed patients have developmental deficits in all three emotions, or if it is simply 5-HT dysfunction. This paper will review a large number of studies, including the genetic and epigenetic aspects of depression pathogenesis, various causative factors such as various developmental stages, and observed alterations in electrophysiological and functional magnetic resonance of the brain. In short, here we try to investigate the neural mechanisms of emotional alterations as well as monoamine neurotransmitters in MDD patients. 

## 3. Monoamine in the Digestive System 

The intestine, also known as the second brain, has a close connection with the brain [55]. Tremendous progress has been made in studying the bidirectional interactions between the gastrointestinal tract and the central nervous system [56], and understanding the interaction between the digestive system with the brain may provide novel approaches for prevention and treatment of MDD [57,58]. Indeed, the digestive system has also been found to play an important role in MDD, including liver [59,60] and intestine [61].

### 3.1. Monoamine and the Microorganisms in the Digestive System

In the digestive system, the increase of serotonin in the gastrointestinal tract leads to aversive nausea and vomiting reactions [62]. Recent studies found that gut microorganisms can activate enteroendocrine cells and produce corresponding hormones such as 5-hydroxytryptamine (5-HT), dopamine (DA), and norepinephrine (NE), which can affect the central nervous system [61]. The brain in turn can regulate gastrointestinal functions through the neuro-immune-endocrine system. At the same time, gut microbes and their metabolites can stimulate the enteric nervous system by regulating intestinal motility and participating in bidirectional gut–brain interactions [61]. During MDD, altered gut microbes modulate the brain mainly through the vagus nerve, the hypothalamic-pituitary-adrenal (HPA) axis, the hypothalamic-pituitary-gonadal (HPG) axis, the immune system, and metabolic pathways, thus causing the hippocampus to preside over mood, emotion, and cognitive functions, which may lead to MDD. Since gut microbes can regulate human emotions, learning, memory, social interaction, ingestion, and other behaviors, the related effects are called gut microbial regulation [61].

### 3.2. Brain–Gut Axis 

The concept of the gut–brain axis emerged from the recent demonstration that the gut microbiota affect depressant behavior [63], and, recently, many scientists proposed that the brain–gut axis is a reason for MDD [64,65,66,67]. For example, Zhang et al. transplanted the gut microbiome of depressed patients into germ-free mice, and induced depression-like behaviors in the recipient mice, suggesting that alterations in the gut flora can induce depressive symptoms [68]. The human intestinal flora is a dynamic process in different stages of life, and adolescents are in the peak of growth and development, whose intestinal microorganisms are in an unstable state and more diverse. It is found that a surge of anaerobic bacteria in the gut flora of adolescents may be associated with the development of mood disorders such as anxiety and depression during adolescence [69]. 

However, no unifying hypothesis has been put forward to explain the physiologic significance of this remarkable phenomenon. Therefore, it is important to investigate the molecular mechanisms of the gut–brain axis interaction and seek new ways to regulate the function of the gut–brain axis for the treatment of adolescent depression. The intestines are closely connected with the brain and also in emotional regulation [70]. Disgust is a basic emotion induced by unpleasant and repulsive stimuli in the gut. Disgust emotion can induce behavior inhibition [49], warding off potential diseases and staying away from or avoiding spoiled food or other contaminants [14,71]. The brain–gut-axis induced disgust might be the reason for MDD; indeed, the brain–gut-axis affects disgust (sad) emotions via two main pathways [61]: (1) Metabolite pathways: Gut microbes can activate monoamines via stimulating the enteroendocrine cells to produce the 5-HT [61]. (2) Neuroendocrine pathway: The intestinal flora affect the monoamine system in the central nervous system by regulating the secretion of these neurotransmitters and hormones, such as endorphin, tryptophan, and 5-hydroxytryptamine. However, the brain–gut-axis also affects the vagal pathway: The enteric nerve forms synaptic connections with the vagus nerve, which composes the parasympathetic nervous system. The parasympathetic nervous system has been known to reduce the emotional arousal, known as the pleasure nerves [72]. 

### 3.3. Brain–Liver Axis

Chinese medicine has a long history related to believing that depression is due to inhibition of anger in the liver, and antidepressant medications in Chinese medicine often have a liver-regulating effect, such as the powder of bupleurum [73]. Recently, it has been found that patients with MDD had a significantly higher prevalence and incidence of chronic liver disease than the general population [74]; thus, the integrative activity of liver and brain might play a role in depression [59,60]. Part of the reason might be due to the metabolites, such as the bile acids [55]. Bile acids are steroid acids which are synthesized in the liver and further processed in the gut by digestive enzymes. Bile acids participate in a range of important host functions such as monoamine neurotransmitters, short-chain fatty acids, and indoles [75]. Non-alcoholic fatty liver disease has complex pathogenic mechanisms for MDD [76]. In addition, the liver is involved in the metabolism of many stress hormones, and neurotransmitters, for example, Cytochrome P450 (CYP), which is primarily expressed in the liver, are some of the main metabolizers of glucocorticoids and monoamines in the peripheral system [77]. Thus, the link between hepatic glucocorticoid metabolism and central monoamine transmission might be important in pathophysiology of stress-related disorders [77]. 

Previously, we proposed that NE and glucocorticoid and sympathetic system might be the neurotransmitter for anger and fear. Lazarus argues that a different expression of fear or anger in a given stressful situation is dependent on the cognitive appraisal of the situation [78]. He proposed that an individual will have the fear emotion, if his primary appraisal is having insufficient resources in a threatening situation; if he feels like he has sufficient resources, the individual will show the anger emotion [79]). However, this cognitive appraisal can be trained, such as for animals in CUMS (chronic unexpected mild stress) training. When the animals are trained at the early stage of CUMS training, they believe they are competent and show anger responses. However, with the training that has progressed for a long time, they are unsure about their competence, show no angry responses, and enter a stage of helplessness, which is a characteristic feature for MDD. Of note, this might be the reason why vulnerable groups, such as adolescents, the elderly, and women, are prone to depression because they can easily become helpless and are afraid to show angry emotions.

## 4. Monoamines and Astrocyte in the Brain

The monoamine hypothesis of MDD provides a robust theoretical framework, forming the core of a large jigsaw puzzle, around which we must look for the vital missing pieces [2]. Recent emphasis on astrocyte-centric cause of MDD has gained more attention during the past several decades [80], and a growing number of papers suggest that astrocytes play a key role in depression [2]. Astrocytes are the most abundant and versatile cells in the brain, and are involved in most brain functions acting as a passive housekeeper or as an active player [81,82]. Astrocytes were considered passive supporting cells, merely maintaining homeostasis by supplying nutrients and cleaning the metabolites [83,84]. Physiological studies during the past decades have found that astrocytes can actively work with these functions by monoamine induced Ca^2+^ signaling [85]. It is found that astrocytes can actively work together in responses to monoamine neurotransmitters, glutamate, and purinergic bases [86,87]. 

Indeed, it is found that monoamine neurotransmitters and most antidepressants have been proved to work through astrocytes [82]. For example, we showed that the synchronized release of norepinephrine (NE) from locus coeruleus (LC) projections throughout the cerebral cortex mediate long-ranging Ca^2+^ signals by activation of astrocytic α1-adrenergic receptors [88]. We also found that astrocytes appear to be the primary target for NE via triggering astrocytic Ca^2+^ signaling through α1-adrenergic receptors; instead, astrocytes do not respond directly to glutamatergic signaling evoked by sensory stimulation; thus, astrocytes may coordinate the broad effects of neuromodulators on neuronal activity [88,89].

### 4.1. Astrocytic Loss Might Be a Reason for MDD

Unlike other diseases (e.g., brain injury, Alzheimer’s disease, Parkinson’s disease, etc.) where astrocytic activity is increased and GFAP (glial fibrillary acidic protein) proliferation (astrogliosis) is predominant, MDD patients exhibit reduced numbers and smaller size. Growing evidence reported astrocyte loss in key regions of the limbic system in depressed patients, which might account for the pathology of MDD [2]. Many studies in animal models have subsequently hinted at the possibility that astrocytic atrophy may play a causative role in the precipitation of depressive symptoms [82]. This reduction in cell number occurring first in astrocyte cells before that in neurons suggests that astrocyte lesions may be responsible for neuronal damage. This raises an intriguing possibility that the astrocytes may play a central role alongside neurons in the behavioral effects of antidepressant drugs. 

### 4.2. Dysfunction of Buffering Ability in Astrocytes 

The involvement of astrocytes in depression might also be due to its effects in buffering extracellular K^+^, as has been demonstrated, for example, in the study published in Nature by Hailan Hu’s laboratory (Cui et al, 2018). It was found that astrocytes in depressed mice have a reduced potassium buffering capacity and that astrocytes act as a ‘potassium reservoir’ in the perineuronal environment to maintain ion homeostasis of the perineuronal environment [90]. The most important function of astrocytes is to exchange nutrients and metabolites through the blood–brain barrier formed by the tight link between their terminal feet and the vascular epithelium, which can be called the glymphatic system [91]. In addition, antidepressants and monoamine neurotransmitters exert profound effects on the gene expression in astrocytes via releasing TGF-β. The inflammatory response induced by our previous experimental stress can trigger astrocytes to secrete some neuroactive substances such as TGF-β. We propose that TGF-β can lead to increased expression and depolarized distribution of NKCC1 and AQP4 in astrocytes, resulting in post-stress brain injury and depression [92]. 

### 4.3. Dysfunction of Glymphatic System 

The glymphatic system is a recently defined system in the brain by our lab, to characterize the brain-wide paravascular pathway for cerebrospinal fluid (CSF) and interstitial fluid (ISF) exchange, which helps efficient clearance of solutes and waste from the brain [93]. Interstitial fluid was formed by cerebrospinal fluid that entered the brain along para-arterial channels, and the interstitial fluid was then cleaned from the brain via the para-venous pathways [94]. The normal function of glymphatic circulation depends on normal function of astrocytic activity, especially the polarized expression of aquaporin-4 (AQP4), which is also called the water channel. Many recent studies found that dysfunction of glymphatic transport, which can be induced by ageing or inflammation, is involved in MDD [95]. Liu et al reported that CUMS can induce depression-like symptoms by decreasing monoamines’ neurotransmitter concentration [96]. Moreover, CUMS decreased polarized expression of AQP4 and inhibited the glymphatic system. Some studies confirmed the "glymphatic dysfunction" hypothesis for MDD, which hypothesized that the stress induced dysfunctional glymphatic pathway serves as a bridge between monoamine disturbance and emotional disorders [94,97]. Indeed, it is found that NE can block the glymphatic system, which works more efficiently during sleep [98]. This is consistent with the idea that there are sleep problems for MDD patients; however, the role of basic emotion anger or NE is not clear.

## 5. Monoamines and Microglial in the Brain

Major depressive disorder (MDD) has been recently suggested to be related to inflammation [99]. In addition, many studies found that stress (fear and anger), which is an important core emotion as we previously suggested [31], really affects inflammation in MDD [100,101]. In addition, it is found that anger can induce cortisol while fear can induce inflammatory cytokine but a decrease in cortisol, which suggested that anger might mobilize energy and trigger adaptive processes, while fear promotes withdrawal behavior [102]. Thus, a lack in anger might be the reason for depression related inflammation, and indeed accumulating evidence started to reveal that stress related emotions might affect monoamine oxidase activity, and activate the primary immune cell microglia [95].

Many recent studies found that impairment of the microglia can lead to depression, and some forms of depression can be considered as a microglial disease (microgliopathy) [103]. Microglia activation induces releases of many inflammatory factors, such as IL-1β, tumor necrosis factor (TNF-α), and transforming growth factor (TGF-β). These inflammation factors are thought to be important factors for MDD [99]) by affecting the function of monoamine neurons. Studies suggested that stress related emotions, especially anger [104], can significantly elevate inflammatory factors, which can induce decreased monoaminergic neurotransmission via oxidative stress. Thus, abnormal function of astrocytes and microglia in the brain is also a major etiology of depression [105]. 

### 5.1. Microglia Cells

Microglia are the resident immune cells, and can induce release of proinflammatory cytokines to induce neuro-inflammation in the central nervous system. Microglia are not resting under physiological conditions; they play an important function in pruning synapses by actively monitoring the development and maturation of synapses. In stressful situations, they can be activated by ATP released from activated astrocytes, in order to induce neuroinflammation [106]. Microglia were activated by ATP to change its morphology with multiple protrusions, with highly branched or amoeboid morphology, and changed their function by releasing cytokines. Neuroinflammation and oxidative stress may provide the impetus for increased transcription of monoamine oxidase [107]; monoaminergic drugs can block both activation of microglia and reverse the depression in some way [108]. 

### 5.2. Early Life Stress Induces Microglia Vulnerability

Early life stress has been suggested to the major reason for MDD [109]. Early life stress induced inflammatory damage that can lead to abnormal neuronal plasticity via enhanced microglia activity and excessive synaptic pruning [110]. Many studies found that early life stress can induce microglia vulnerability in later lives [111], by altering phagocytic activity, gene expression, and morphology [109]. Consistently, it has been shown that chronic stress induces depressive symptoms by affecting microglia’s pruning ability and synaptic plasticity [112].

Microglia and neuronal interactions play important roles in synaptic plasticity and mediated behaviors during adolescence. Stressful lives during adolescence can induce rapid microglia changes in distribution, morphology, and function, significantly affecting neuronal and synaptic development [113]. Strong stress can induce inflammatory responses by releasing many cytokines, such as IL-6 and TNF-α, which are activated by enhanced expression of some proteins, such as the transcriptional fingerprint, especially the NF-κB pathway—together with reduced expression of other proteins, such as glucocorticoid receptor expression.

### 5.3. Acute Stress

Normal activation of microglia protects neurons by performing synaptic shearing and phagocytose excess proteins, but excessive neuroinflammatory response is neurotoxic and harmful. After acute stress, microglia are slowly activated by ATP (or excessive glutamate) via P2X7 receptors, which enhance intracellular Iba-1, CD68 expression, and enhanced expression of OX-42, RCA-21 etc., and resulted in activated inflammatory vesicles and the release of more inflammatory factors [114]. In addition, microglia release many miRNAs through exosomes [115]. Exosomes are structures to spread information, especially disease information like cancer and inflammation, by cellular excretion of proteins, mRNAs, and miRNAs in exosomes. It has been reported that stress responses (e.g., heat shock, and oxidative stress) can induce exosome secretion. miRNA expression in MDD patients is in an abnormally active state and can interfere with the normal function of glial cells and neurons [116]. Many miRNAs have been proven to play an important role in MDD pathogenesis and were suggested to be biomarkers for MDD, such as miR-1202. It follows that, if chronic stress is encountered, microglia undergo an inflammatory response and will also release many miRNAs through exosomes, thus interfering with the normal function of astrocytes and neurons [117]. In addition, inflammatory molecules can affect the developing brain by influencing the hypothalamus-pituitary-adrenal axis (HPA), glial cells, and monoamine metabolism [118].

## 6. Monoamines and Neuroendocrine

Inflammatory cytokines can modulate mood behavior and cognition by affecting brain monoamine levels, activating neuroendocrine responses, promoting excitotoxicity (increased glutamate levels), and impairing brain plasticity [119]. Monoaminergic modulation of emotion relies on both the nervous system and neuroendocrine. The major structure for neuroendocrine is the hypothalamus-pituitary-adrenal axis (HPA). HPA works as a very important complementary and executive system for the emotional function of the monoamine system. For example, cortisol and the adrenocorticotropin-releasing hormone (ACTH) are important neurotransmitters during angry responses, assisting in sympathetic regulation and emotion-related physiological functions [120]. In addition, the hypothalamic-pituitary-gonadal axis (HPG axis), which regulates estrogen and progesterone, has a very strong influence on aversive mood and may be the main reason for the high prevalence of depression in women [121]. In addition, oxytocin released by the hypothalamus is considered to be the neurotransmitter of love and attachment and plays a very important regulatory role in the recognition of facial expressions [122,123].

### 6.1. Hypothalamic-Pituitary-Adrenal (HPA)

The hypothalamic-pituitary-adrenal (HPA) axis is an important component of the neuroendocrine system and is involved in the regulation of the body’s stress response. Under physiological conditions, glucocorticoid (GC) negatively feeds back to the brain by regulating the release of monoamine neurotransmitters, inhibiting the activity of HPA, and weakening the stress response. However, when stressful stimuli persist, consistent high levels of glucocorticoids can lead to the abnormal release of monoamine neurotransmitters in the brain [124]. Glucocorticoid receptors (GR) and GR mRNA expression are reduced in peripheral blood and brain tissue of depressed patients, and antidepressant treatment ameliorates abnormalities in GR. Co-chaperone proteins of GR play important roles in normal folding, maturation, nuclear translocation, and binding of GR to DNA. Among them, FK506 binding protein 51 (FKBP51) is an important co-chaperone protein [125].

Enhanced FKBP5 transcription and translation reduce the sensitivity of GR. FKBP5 knockout mice had lower adrenal and thymus weights and corticosterone levels after chronic stress swimming for a longer time in the forced swimming test. In contrast, FKBP5 knockout mice showed an age-dependent antidepressant-like phenotype with enhanced stress resilience. Elevated basal FKBP5 levels in depressed patients can be a predictor of depressive disorders, while genetic polymorphisms in FKBP5 interact with early-life stress to increase the risk of developing depressive disorders. In terms of response to treatment, studies suggest that FKBP5 can be used as an independent factor to predict responsiveness to treatment in depressed patients [126]. It is found that hippocampal and prefrontal GR expression was reduced, and FKBP5 was elevated in adolescent chronic stress depression model animals. It is suggested that GR/FKBP5 might be a biomarker for MDD, for it is a key factor in the development and increased susceptibility to depression, and jointly mediate HPA axis function and stress response [127].

Regarding the regulation of FKBP5 expression, miR-511 was found to decrease the protein level of FKBP5 and downregulate the glucocorticoid-induced upregulation of FKBP5. In contrast, miR-511 overexpression significantly enhanced GR activity. Chronic stress experiences during adolescence lead to reduced miR-511 expression in the prefrontal lobe of adult animals. Functionally, FKBP5 might also be a biomarker because of its properties to regulate GR, such as regulating pathways related to immune function, autophagy, epigenetic remodeling, apoptosis, cell growth, cytoskeleton dynamics, and metabolism. In addition, miR-511 contributes to the age-dependent regulation of FKBP5 and neuronal development and differentiation [128].

### 6.2. Hypothalamic-Pituitary-Gonadal Axis (HPG)

Estrogen and several other sex hormones have been proved to affect emotions and play a role in MDD, which might be a major reason for high prevalence of MDD in women [129]. The pathogenesis of pre-perimenopausal depression is also suggested to be related to reduced levels of estrogen (estradiol, E2), and E2 replacement therapy has been used to treat perimenopausal depression [130]. However, recent studies have found that E2 has little effect on perimenopausal depression; instead, estrogen deficiency-induced female depression might be related to 5-hydroxytriptamine (5-HT) deficiency [36]. In addition, luteinizing hormone (LH) is significantly altered during the menstrual cycle, perinatal, and perimenopausal periods, which may be the main cause of depression [131]. Moreover, increased LH can enhance the release of cortisol and lead to Cushing’s syndrome and depression. In addition, depression is one of the typical symptoms of Cushing’s syndrome [132]. The main function of LH receptors is to promote glandular proliferation and hormone (transmitter) secretion. LH can activate the PI3K-Akt-mTOR pathway in ovarian epithelial cells and granulosa cells, increasing the phosphorylation of mTOR substrates S6K1 and EIF4EBP1 to promote hormone synthesis in the corpus luteum [133].

LH changes dramatically during the menstrual cycle, perinatal period, and perimenopause. LH secretion before ovulation can reach 3–8 times the basal level, and LH secretion increases to three times the previous level during perimenopause. During perimenopause, ovarian function gradually deteriorates and the ability to secrete E2 and progesterone decreases. The hypothalamic-pituitary-gonadal axis (HPG) is dysfunctional. Clinical studies suggested that persistent LH elevation can induce the clinical ACTH Independent Cushing’s Syndrome (ACTH) [134]. Recently, it has been reported that Cushing’s syndrome patients with increased cortisol secretion exhibit reduced mTOR function while inhibition of the mTOR pathway on the adrenal glands can reduce cortisol secretion. LH can activate the mTOR pathway on uterine membrane cells and granulosa cells, which leads to estrogen secretion. Studies also found a synchronous and persistent elevation of LH and cortisol in ovariectomized rats in a stressful environment, and we hypothesized that LH could induce cortisol secretion through the mTOR pathway [135]. Furthermore, the persistent increase in ACTH and cortisol secretion is an important predisposing factor for depression [136]. Therefore, increased peripheral LH secretion might be a reason for perimenopause MDD by inducing cortisol activity, which can lead to depression.

## 7. Neurotrophic, Nerve Regeneration, and Neuroplasticity

Major depressive disorder (MDD) is a common emotional disorder with etiological heterogeneity and complex molecular mechanisms which are not fully understood. Recent studies in postmortem or animals have shown that over-activated microglia can inhibit neurogenesis and induce MDD [137]. It is also found that neurotrophic/growth factors such as the brain-derived neurotrophic factor are decreased in postmortem brains from suicide victims, which suggests that altered trophic support might be a reason for the pathophysiology of MDD [138]. BDNF has recently been regarded as an interesting target for developing new antidepressant drugs [139]. 

In realizing the limitations of current antidepressant therapy, depression research has branched out to encompass other areas such as synaptic plasticity, neurogenesis, and brain structural remodeling as factors which influence mood and behavior. Monoamine neurons, their neural loops, and neural connections are in a lifelong process of reorganization and modification. Studies suggest that neuroplasticity is involved in the pathogenesis of depression. Synaptic plasticity includes structural plasticity and functional plasticity, which modulates the function of local neuronal networks or corresponding neural loops by altering normal neural firing, neurosecretion, and other activities by affecting the information transfer between neurons, and plasticity modulation targeting the corresponding neural loops will be the urgent need and main direction of research in these psychiatric disorders [140]. 

### 7.1. BDNF

Reduced neuroplasticity is an important pathophysiological basis for psychiatric disorders including depression. Brain-derived neurotrophic factor (BDNF) is an important mediator involved in the regulation of normal brain development and neuroplasticity [141], and indeed BDNF abnormalities in signaling pathways play an important role in the development of depression [142]. BDNF gene polymorphisms may be risk factors for the development of depression and interact with the genetic environment to play a role in the development of depression. Based on the established database of comprehensive evaluation characteristics of adolescent bipolar demographics, genotyping, and behavior at the Institute of Psychology, Chinese Academy of Sciences, a bipolar design was used to isolate and identify the extent to which genetic, shared environmental, and nonshared factors influence adolescent mood by developing structural equation models [143]. The results revealed that the main effect of BDNF val66met on adolescent depression was not significant, which means that the allele present at the polymorphic locus did not predict depression. However, the BDNF val66met together with stressful life events interaction was significant, suggesting that adolescent individuals with different alleles at this polymorphic locus reacted differently after exposure to stressful life events [144]. In-depth analysis revealed that individuals with the Val allele were prone to depression after exposure to high levels of stressful life events relative to individuals who were met/met pure congenic. However, in the absence of stressful life events or at very low levels, Val allele carriers were again protected, which indicates that they were less prone to depression [145]. Furthermore, previous gene–environment interactions have failed to control well for confounding gene-environment correlations in study results. To better rule out such confusion, a bipartite design was used to isolate the part of the environment that is not genetically influenced as a pure environmental factor and observed the effect of the BDNF val66met polymorphic locus on depression by interacting with these pure environments. The findings confirmed previous findings that Val allele carriers are more sensitive to environmental factors in the Chinese population and are more prone to depression than met/met pure congeners at higher levels of stressful life events [146]; thus, BDNF and its related modulations might be a biomarker for MDD.

### 7.2. Epigenetics

Epigenetic mechanisms are important mechanisms for translating the effects of environmental factors into specific patterns of gene expression and can affect brain circuits over time [147]. The influences of early social environmental on animal and human behavior are associated with alterations in epigenetic modifications [148]. It is found that depressive chronic social stress continues to affect animals’ depressive behavior in adulthood, including deficits in social interest, impairments in executive function, and so on [149]. Epigenetic modifications in the medial prefrontal cortex BDNF gene are altered, primarily by elevated levels of histone H3K9 methylation (a transcriptional repressor of BDNF) in regions downstream from this promoter region, leading to an expression, mediating the long-term effects of adolescent depression on behavior and brain development [150].

## 8. Monoamine Neurotransmitter and Other Neurotransmitters

Psychological analyses have made a lot of progress in emotional studies, and these pioneering works have helped locate many important structures involved in basic emotions. However, the characteristic feature of monoamine neurotransmitters is their broad projection, which can affect the function of the whole brain and even the whole body [151]. Thus, monoamine neurotransmitters are also called neuromodulators, including dopamine, norepinephrine, and serotonin (DA, NE, and 5-HT). These monoamine neuromodulators might be the primary neural basis for emotions, and we suggested that emotions are nothing but neuromodulators [15,16,35].

The monoamine neuromodulators are the primary neural basis for emotions; however, there are many other chemicals in the brain that are related to emotions, such as corticotropin-releasing hormone (CRH) or cortisone, or oxytocin, or sex hormone. All these neurotransmitters, their mutual interaction and their influence on other processes, as stated before, might be different in individual patients; therefore, there might be many chemical dysfunctions that induce clinical heterogeneity among MDD patients. In this context, all chemical changes might play a role of biomarkers for the diagnosis of MDD. Overall, the monoamine neuromodulators might be the primary neural basis for emotions, and the other related chemicals, including the hormones, inflammation cytokines, neuro-tropical factors, or mRNA, LnRNA might work as biomarkers for MDD (Figure 2).

There are many chemicals in the brain that are related to emotions; we cannot review them in this paper due to the paper’s length. We can use GABA as one example to introduce the interactions between monoamine and other chemicals. The specific neurotransmitters include glutamatergic excitatory neurons and GABAergic inhibitory interneurons, which are the main executive systems regulating neural circuit connections in emotional brain regions and between brain regions [152]. The structure and function of glutamatergic and GABAergic neurons in the prefrontal cortex and hippocampus of brain regions and the imbalance of their local loop activity are also the monoaminergic targets for the antidepressant drug treatments [153]. In addition, glutamatergic and GABAergic neurons can in turn counteract monoamine neurons; for example, direct or indirect excitatory neural projection from the prefrontal lobe can induce top-down emotional and cognitive regulation to the limbic system [154].

The interactions between glutamatergic and GABAergic systems can be affected by negative life events at early childhood, such as a mother–infant relationship and adolescent social stress, and results in emotional behavior [75]. GABAergic activates depending on ionotropic GABA_A_ and metabotropic GABA_B_ receptors. Some drugs targeting the GABA system, including GABA_B_ receptor modulators, such as CGP36742, CGP 51176, and CGP56433, and GABAA receptor α2 and α5 subunit orthosteric agents, have been reported to have rapid antidepressant effects. On the other hand, NMDA receptors might also be a target for rapid antidepressant research, such as ketamine, which has been used as an antagonist for NMDA receptors. In addition, some other agonists or enhancers targeting the Glycine Binding Site (GBS) site on the NMDA receptor have also shown antidepressant effects [82]. However, their interaction with monoamine activity is not clear. Oxytocin has been recently found to be the neuromodulator for love and attachment [155]. Nasal application of oxytocin can induce positive emotions. In addition, the hormones from the hypothalamic-pituitary-adrenal (HPA) axis, such as CRH, ACTH (adrenocorticotropic hormone), and cortisol, have been proved to be involved in the stress process. Furthermore, the sex hormone progesterone has been proved to affect disgust [35]. Overall, many neurotransmitters have been suggested to be involved in the emotions.

## 9. Conclusions

The monoamine hypothesis has been largely supported by the pharmaceuticals that target monoamine neurotransmitters as a treatment for depression. Therefore, the first-line antidepressant drugs remain for raising monoamine neurotransmitters. However, these antidepressants have come under scrutiny due to their limited clinical efficacy, side effects, and delayed onset of action [156]. There is increasing evidence that glial cells play a role in the pathological mechanisms of mood disorders and the mode of action of antidepressant drugs. Thus, the new etiological hypotheses of MDD should extend the monoamine transmitter hypothesis to a broad area including many body systems and many kinds of brain cells, in order to clarify the main hypothesis for the pathogenesis of depression.

## 10. Future Perspectives:

This paper advanced the monoamine hypothesis for basic emotions and MDD; however, there are still some limitations for this review. MDD is quite a big field of study, and there are lots of publications every year; thus, we cannot give a full review of the literature. For example, we did not review the large amount of material about biomarkers for MDD, which is a very interesting approach for MMD diagnosis for precision medicine (especially for psychiatry). The interactions among the neurotransmitters involved in MMD, and their mutual interaction and their influence on other processes, as stated before, are very different in individual patients; therefore, it is critically important to screen the clinical heterogeneity among MDD patients, which will facilitate decision-making by introducing measurable biomarkers (i.e., indicators of normal biological processes, pathogenic processes, or responses to the exposure or intervention). This field of precision medicine will make tremendous progress in science and technology, enabling the acquisition and processing of enormous amounts of information and identification of associations between individual patient’s characteristics, as well as diagnosis, prognosis, and treatment methods for MDD. 

## Figures and Tables

**Figure 1 pharmaceuticals-15-01203-f001:**
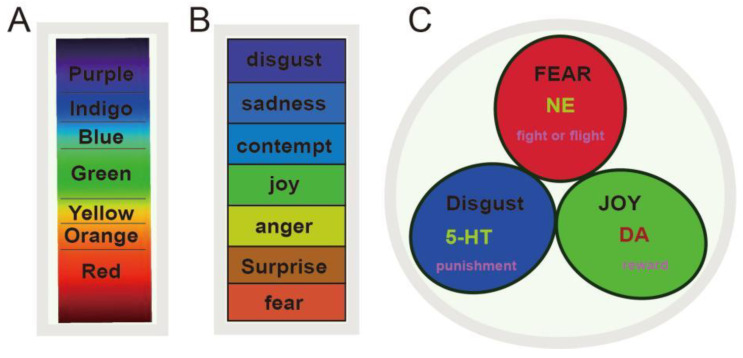
The primary color model of basic emotions. (**A**). Newton’s prismatic colors diagram, and Newton says there are seven primary colors. (**B**). The picture shows Ekman’s seven basic emotions (fear, anger, joy, surprise, contempt, sadness, and disguZ`st). (**C**). The picture shows our hypothesis of three primary emotions: joy, disgust (sadness), and fear (anger), which are linked to three monoamine neurotransmitters.

**Figure 2 pharmaceuticals-15-01203-f002:**
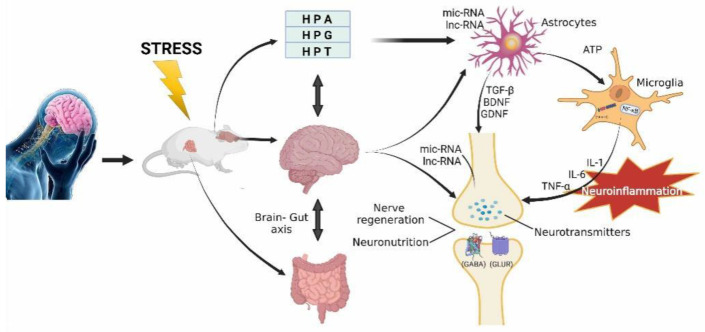
The mechanisms of MDD. Pharmaceuticals have greatly supported the monoamine hypothesis by their efficient effects on the monoamine neurotransmitters. Their new effects should be investigated in many different systems, including neuroendocrine and digestive system, and also in many kinds of cells in the brain, including neurons and glia. Thus, the new mechanisms should be extended to neurotrophic, neuro-inflammation, metabolites as well as nerve regeneration.

## Data Availability

Data sharing not applicable.

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
