# Peer review of "Monoamine Neurotransmitters Control Basic Emotions and Affect Major Depressive Disorders"

_pharmaceuticals, 2022, doi:10.3390/ph15101203_

Round 1

Reviewer 1 Report

Pharmaceuticals is an international scientific journal of medicinal chemistry and related drug sciences. Further, Pharmaceuticals is a multidisciplinary journal. So, the spectrum of readers  is broad.

The paper is interesting, summarizing hypotheses dealing with neurotransmitters.

 I have several suggestions:

 1. At the beginning of the paper, a  brief and  clear  explanation how the three primary colour model of basic emotions fits into other hypothesis focused on interaction of neurotransmitters and other systems should be included.

2. To add  a discussion

- why do authors prefer the hypothesis  dealing with  microglia?

-  the clinical heterogeneity that exists among patients limits the ability of MDD to be accurately diagnosed;  all neurotransmitters,  their mutual interaction and their influence on other processes are important less or more in individual patients. In this context a role of available biomarkers should be mentioned.

3. Conclusion

To add a vision for future     ……. role of precision medicine (psychiatry), which  facilitates decision-making by introducing measurable biomarkers ( i.e., indicators of normal biological processes, pathogenic processes, or responses to the exposure or intervention)……. A tremendous progress in science and technology enables the acquisition and processing of enormous amounts of information and identification of associations between individual patients’ characteristics, and diagnosis, prognosis and treatment methods and success on the other.

Reviewer 2 Report

The review is titled “Monoamine Neurotransmitters Control Emotional Arousal and Affect Major Depressive Disorders via Glial Cells” but only sections 3 and 4 discuss the role of glia in MDD-related monoamine abnormalities, so I suggest that this review should have a broader title.

No references appear in many sentences, especially in "our previous data", and even paragraphs. For example: section 5.1 (second and third paragraph) and 5.2 (second paragraph). Please, revise the text to include appropriate references when citing data.

I strongly suggest that the third paragraph of section 1.3 should be removed for several reasons. First there is already a section 2.3 where the possible role of the liver in depression is mentioned and some sentences are very speculative and controversial. For example: “lacking anger may be an emotional etiology of depression” or “Of note, this might be the reason why vulnerable groups, such as adolescents, the elder, and women, are prone to depression, because they are easy to be helpless, and afraid to show anger emotions.”

In section 1.4, I also suggest change the phrase “but the relationship between serotonin (5-HT) and depression is still unclear”. There is an amount of papers showing the relation of the serotoninergic system and MDD (Artigas, F., 2013. Pharmacol Ther 137, 119-131;  De Vry, et al 2004. Eur Neuropsychopharmacol 14, 487-495).

Figure 2 is not referenced in the text.

Section 1 should appear in bold

Reviewer 3 Report

Review: Monoamine Neurotransmitters Control Emotional Arousal and Affect Major Depressive Disorders via Glial Cells

In this review, Yao Jiang et al firstly introduce basic concept of Major Depressive disorders, then they move on to discuss the relationship between the  MDD and the monoamine neurotransmitters such as DA, NE, and 5-HT . They give a complete picture about the Three Primary Color Model of Basic Emotions (joy, fear and disgust). They also explain the contribution of the monoamine in many side such the digestive tract, astrocytes, microglial and other. Finally, they very briefly address the potential of Monoamine Neurotransmitters as therapeutic approach for MDD patients even his limited clinical efficacy, side effects, and delayed onset of action.

The work is well organized and comprehensively described, correct and readable.
